# Analysis of Relationship between Microwave Magnetic Properties and Magnetic Structure of Permalloy Films

**DOI:** 10.3390/s24196165

**Published:** 2024-09-24

**Authors:** Nikita A. Buznikov, Andrey N. Lagarkov, Sergey A. Maklakov, Sergey S. Maklakov, Alexey V. Osipov, Konstantin N. Rozanov, Polina A. Zezyulina

**Affiliations:** Institute for Theoretical and Applied Electromagnetics, Russian Academy of Sciences, Moscow 125412, Russia; n.a.buznikov@gmail.com (A.N.L.); sergeymaklakov@yandex.ru (S.A.M.); squirrel498@gmail.com (S.S.M.); avosipov@mail.ru (A.V.O.); k.rozanov@yandex.ru (K.N.R.); zez-p@yandex.ru (P.A.Z.)

**Keywords:** permalloy films, microwave permeability, ferromagnetic resonance, out-of-plane anisotropy, stripe domain structure

## Abstract

Changes in the microwave permeability of permalloy films with an increase in the film thickness are studied. Measurement data on the evolution of microwave permeability with film thickness are analyzed in the framework of a model for the film with a regular stripe domain structure and out-of-plane magnetic anisotropy. A correlation between the microwave magnetic properties and magnetic structure of permalloy films is established. It is demonstrated that the observed decrease in the ferromagnetic resonance frequency and the static permeability with a growth in the film thickness can ascribed to the appearance of perpendicular anisotropy and the formation of a stripe domain structure. The calculated dependences of the ferromagnetic resonance frequency and static permeability on the film thickness are in reasonable agreement with the measurement results. Based on the analysis of these dependences, the domain width in the permalloy films is estimated. It is found that for thick permalloy films, the domain width is of the order of the film thickness. The results obtained may be useful for high-frequency applications of soft magnetic films.

## 1. Introduction

In recent decades, thin soft magnetic films have been studied extensively due to their possible use in high-frequency applications, such as magnetic inductors [1,2,3,4,5,6,7,8,9,10], patch antennas [7,8], micro-transformers [3], magnetoelastic sensors [2,9], tunable microwave filters [5,6,10], electromagnetic shielding [11], etc. In these applications, high values of microwave permeability are required. The microwave permeability of a material may be estimated based on the static permeability *μ*_s_ and the ferromagnetic resonance frequency *f*_res_. To achieve high values of the microwave permeability, both *μ*_s_ and *f*_res_ should be as high as possible.

For bulk magnetic materials, the product of these values is limited by the Snoek law [12]. Soft magnetic films are promising for obtaining high microwave permeability, since the product of the static permeability and ferromagnetic resonance frequency in the films may exceed the Snoek limit [13,14,15]. For thin magnetic films, the microwave permeability is evaluated by using the Acher parameter:(1)$$k_{A}=\frac{(\mu_{s}-1)f_{\mathrm{res}}^{2}}{{\left(\gamma 4\pi M_{0}\right)}^{2}},$$
where *M*_0_ is the saturation magnetization of the material and *γ* ≈ 3 GHz/kOe is the gyromagnetic ratio.

For a uniformly magnetized film with in-plane magnetic anisotropy, *k*_A_ tends to unity. In real magnetic films, the Acher parameter may decrease (*k*_A_ < 1) as a result of deviations from the uniform magnetic structure, in particular, due to the appearance of out-of-plane magnetic anisotropy and a domain structure [15,16]. The model for a magnetic film having a stripe domain structure and an arbitrary angle between the anisotropy axis and the film plane predicts a decline in microwave permeability [17]. This prediction is in a qualitative agreement with the results of the measurements of microwave permeability in permalloy [18] and cobalt films [19] with different thicknesses.

Permalloy films attract much attention due to their low coercivity, high static permeability and low magnetocrystalline anisotropy, resulting in high microwave permeability [20,21,22,23,24,25]. The excellent soft magnetic properties of permalloy films are very promising for sensors applications, in particular, for the development of giant magnetoimpedance sensors [26,27,28,29]. However, out-of-plane anisotropy can appear in permalloy films in cases when the film thickness exceeds some critical value [20,30,31,32,33,34,35,36,37,38]. The appearance of out-of-plane anisotropy may be related to columnar structure formation as well as magnetocrystalline and magnetoelastic anisotropy. The presence of out-of-plane anisotropy in a film can lead to a transition into a transcritical state. The transcritical state is characterized by a specific hysteresis loop (so-called transcritical loop) [39], enhanced coercivity and the formation of a stripe domain structure in the film. As a result, the microwave permeability of a film is significantly reduced after its transition into a transcritical state.

This paper deals with a quantitative comparison of the results predicted by the model in [17] with measurement data for the microwave permeability of permalloy films. Based on the microwave permeability data, changes in the magnetic structure of permalloy films appearing with an increase in the film thickness are analyzed. It is found that the observed decrease in the static permeability and the ferromagnetic resonance frequency is related to the appearance of perpendicular magnetic anisotropy and formation of a stripe domain structure in the films. The width of the domains in the films is estimated by the analysis of the dependence of the Acher parameter on the film thickness.

## 2. Model

In this section, we briefly describe the main results obtained by using a previously proposed model [17]. It was assumed that the film has uniaxial anisotropy, and the anisotropy axis makes the angle *ψ* with the film plane. It was also assumed that the film has a regular stripe domain structure. The motion of the domain walls was neglected. The geometry and angles used in the model are presented schematically in Figure 1.

Note that in general case, the stripe domain structure arises in films as a result of competition between perpendicular magnetic anisotropy, exchange interactions and magnetostatic energy. It is well known that the stripe domain structure exists if the film thickness exceeds some critical value [40,41,42]. This means that there is no stripe domain structure in very thin ferromagnetic films.

For films with out-of-plane anisotropy, the effect of the stripe domain structure on the demagnetizing factor *N_z_* in the direction transverse to the film plane is described in [43,44,45].
(2)$$N_{z}=\frac{32{\left(1+Q^{-1}\right)}^{1/2}}{\pi q}\times \sum_{j=1}^{\infty }\frac{1}{{\left(2j-1\right)}^{3}}\times \frac{1}{1+{\left(1+Q^{-1}\right)}^{1/2}\coth[(2j-1)q]}$$

Here, *q* = (*πd*/2*a*)(1 + *Q*^−1^)^1/2^; *d* is the film thickness; *a* is the domain width; *Q* = *H_a_*/4*πM*_0_; *H_a_* is the anisotropy field. For very thin films, for which *a*/*d* >> 1, the demagnetizing factor tends to 4*π* as in the case of the single-domain film. The value of *N_z_* decreases monotonically with the domain aspect ratio *a*/*d*, and at *q* >> 1, the demagnetizing factor *N_z_* is expressed as in [17]:(3)$$N_{z}\approx \frac{56\zeta (3)}{\pi^{2}}\times \frac{a/d}{1+{\left(1+Q^{-1}\right)}^{1/2}},$$
where *ζ*(3) ≈ 1.20 is the Riemann *ζ* function of 3.

The equilibrium magnetization angle *θ* with respect to the film plane was found by minimizing the free energy that consists of anisotropy and demagnetizing energy. In general case, the magnetization angle *θ* is given by
(4)$$\tan2\theta =\frac{(4\pi Q/N_{z})\sin2\psi }{1+(4\pi Q/N_{z})\cos2\psi }.$$

It should be noted that for soft magnetic films with relatively wide domains, where 4*πQ*/*N_z_* << 1, the equilibrium magnetization angle *θ* is small [17]. In this case, the magnetization deviates slightly from the film plane, and Equation (4) is rewritten as
(5)$$\theta \approx (2\pi Q/N_{z})\sin2\psi \;[1-(4\pi Q/N_{z})\cos2\psi ]$$

The ferromagnetic resonance frequency *f*_res_ was found by solving the linearized Landau–Lifshitz–Gilbert equation. In the framework of the model, the resonance frequency is defined as the frequency where the real part of the permeability *μ*’ = 1. The resonance frequency *f*_res_ depends on the anisotropy axis deviation angle *ψ*, the factor *Q* and the domain aspect ratio *a*/*d* and is expressed as
(6)$$\begin{array}{cc} f_{\mathrm{res}}^{2}/{\left(\gamma 4\pi M_{0}\right)}^{2} & ={\left(N_{z}/4\pi \right)}^{2}\sin^{2}\theta +Q(N_{z}/4\pi )[\cos^{2}\psi -2\sin\psi \sin\theta \cos(\psi -\theta )]\\ & \;+Q^{2}\cos^{2}(\psi -\theta )\;. \end{array}$$

For soft magnetic films, for which *Q* << 1, with relatively wide domains, when *θ* << 1, Equation (6) is simplified as follows:(7)$$f_{\mathrm{res}}^{2}/{\left(\gamma 4\pi M_{0}\right)}^{2}\approx Q\cos^{2}\psi \;[(N_{z}/4\pi )+Q].$$

It follows from Equation (7) that the ferromagnetic resonance frequency *f*_res_ decreases monotonically with a growth in the anisotropy axis angle *ψ* and with a decrease in the domain aspect ratio *a*/*d* [17].

The value of the static permeability *μ*_s_ is found from the general expression for permeability at zero frequency. In general cases, the static permeability is written as
(8)$$\mu_{s}=1+{\left(\gamma 4\pi M_{0}\right)}^{2}[(N_{z}/4\pi )\cos2\theta +Q\cos2(\psi -\theta )]/f_{\mathrm{res}}^{2}.$$

From Equations (1), (6) and (8), the Acher parameter *k*_A_ is given by
(9)$$k_{A}=(N_{z}/4\pi )\cos2\theta +Q\cos2(\psi -\theta ).$$

If the deviation of the equilibrium magnetization angle is small, *θ* << 1, the Acher parameter is simplified as
(10)$$k_{A}\approx (N_{z}/4\pi )+Q\cos2\psi .$$

Since *Q* << 1, the Acher parameter depends slightly on the deviation of the anisotropy axis from the film plane and is mainly governed by the domain aspect ratio *a*/*d*.

## 3. Experimental

Thin permalloy films with a nominal composition of Ni_80_Fe_20_ on a polyethylene terephthalate (PET) substrate were obtained by DC magnetron sputtering at room temperature. The films were sputtered under an Ar flow at 0.67 Pa pressure. A flexible substrate of 12 μm thickness was fixed on a rotating barrel-type substrate holder. The barrel height was 20 cm, while the diameter was 20 cm. A planar extended magnetron sputtering source was used, and the sputtering power per target area was 10 W/cm^2^. The sputtering target-to-substrate distance was 5 cm. The film thickness was controlled by a ZYGO New View 7300 laser interferometer.

The microwave permeability measurements were carried out in a 7/3 coaxial measurement line. A scheme of the measurement setup can be found elsewhere [46,47]. For the measurement, the strips were cut from a film and wound into a hollow cylindrical roll. The inner and outer diameters of the wound stripes were 3 and 7 mm. The frequency dependence of microwave permeability was measured in the range of 0.1 to 5 GHz. The direction of the cut film stripes was chosen to obtain the maximum response in permeability, with the easy magnetization axis being across the strip. The microwave permeability was measured in the absence of an external magnetic field.

## 4. Results and Discussion

The measured frequency dependence of microwave permeability for permalloy films with different thicknesses is shown in Figure 2. It follows from Figure 2 that both the static permeability *μ*_s_ and ferromagnetic resonance frequency *f*_res_ drop sharply if the film thickness exceeds 220 nm. Note that the static permeability was obtained by approximating the experimental data on the real part of the permeability to the low frequency range. The resonance frequency decreases from 1.7 to 0.6 GHz when the film thickness increases from 80 to 1760 nm. The static permeability drops by a factor of four with an increase in the film thickness. A decrease in both the resonance frequency and static permeability leads to a significant decline in the Acher parameter given by Equation (1).

A qualitative explanation for the observed evolution of the ferromagnetic resonance frequency and static permeability with the thickness of permalloy film is as follows. For a film with a thickness of 80 nm, the anisotropy axis is located in the film plane. With an increase in the film thickness, the anisotropy axis deviates from the film plane, and perpendicular anisotropy appears. An increase in perpendicular magnetic anisotropy may be related to columnar microstructure formation and a magnetoelastic effect [37]. A growth in perpendicular anisotropy and a corresponding increase in the anisotropy axis angle *ψ* lead to a decrease in the ferromagnetic resonance frequency *f*_res_ in accordance with Equation (7). It should be noted that the appearance of perpendicular magnetic anisotropy is confirmed by the observed transition to transcritical hysteresis loops in thick permalloy films [18].

Furthermore, an increase in the film thickness is accompanied by a decrease in the static permeability (see Figure 2). This indicates that the in-plane anisotropy field also increases with the film thickness. A change in in-plane anisotropy may be related to the magnetoelastic effect, which contributes greatly to the measured microwave permeability of the films [47]. Note that although the magnetostriction constant in Ni_80_Fe_20_ alloy has a low value [48], the presence of magnetoelastic effects in Ni_80_Fe_20_ thin films obtained by DC magnetron sputtering was reported previously [18,37]. An increase in in-plane anisotropy was also observed in hysteresis loops [18]. An analysis of changes in internal stresses in permalloy films leading to an increase in anisotropy is beyond the scope of this paper. However, it should be noted that the magnetoelastic origin of the increase in anisotropy is confirmed by data on microwave permeability obtained for amorphous films with very low magnetostriction [49]. In these films, a decrease in the microwave permeability was not observed even for relatively thick films due to a low magnetoelastic effect. For further analysis, we introduce the in-plane *H*_in_ and perpendicular *H*_per_ anisotropy fields. Taking into account that *θ* << 1, these fields can be expressed in terms of the effective anisotropy field *H_a_* and the anisotropy axis angle *ψ*:(11)$$H_{\mathrm{in}}=H_{a}\cos\psi \cos(\theta -\psi )\approx H_{a}\cos^{2}\psi ,$$
(12)$$H_{\mathrm{per}}=H_{a}\sin\psi \cos(\theta -\psi )\approx H_{a}\sin\psi \cos\psi .$$

Figure 3 shows the values of the anisotropy axis angle *ψ* and anisotropy fields *H_a_*, *H*_in_ and *H*_per_ obtained as a result of fitting the measured data on microwave permeability. The effective anisotropy field *H_a_* and the anisotropy axis angle *ψ* are obtained from calculations of the resonance frequency *f*_res_ and the static permeability *μ*_s_ by using Equations (2), (4), (7) and (8) and from a comparison of the calculated and measured values of *f*_res_ and *μ*_s_. It follows from Figure 3 that the effective anisotropy field *H_a_* increases sharply when the film thickness exceeds 220 nm and the transition into a transcritical state appears. In this case, the perpendicular anisotropy field *H*_per_ becomes higher than the in-plane anisotropy field *H*_in_.

A comparison of the measured and calculated dependences of the ferromagnetic resonance frequency *f*_res_ and the static permeability *μ*_s_ on the film thickness is shown in Figure 4. For calculations, we use the value of the saturation magnetization *M*_0_ = 805 G and the values of the effective anisotropy field *H_a_* and anisotropy axis angle *ψ* presented in Figure 3. The resonance frequency and the static permeability are found by using Equations (7) and (8), respectively. It follows from Figure 4 that the calculated dependence of the resonance frequency is in good agreement with the results of the measurements, and the discrepancy between the calculated and measured values of static permeability does not exceed 10%.

Figure 5 shows the dependence of the Acher parameter *k*_A_ on the film thickness restored from the results of measurements and calculated using the model. The Acher parameter reduces from 1 to 0.08 with a rise in the permalloy film thickness from 80 to 440 nm. The sharp drop in *k*_A_ is attributed to the transformation in the film magnetic structure with an increase in film thickness leading to the appearance of perpendicular anisotropy and a stripe domain structure.

As mentioned above, in soft magnetic films with out-of-plane anisotropy, the Acher parameter depends mainly on the value of the demagnetized factor *N_z_* (see Equation (10)). According to Equation (2), the value of *N_z_* is governed by the domain aspect ratio *a*/*d*. Therefore, we can estimate the domain width *a* by using the obtained dependence of the Acher parameter on the film thickness.

Shown in Figure 6 is the dependence of the domain width on the film thickness calculated by Equations (2), (4) and (9). The domain width decreases sharply when the film thickness exceeds 220 nm and the transition into a transcritical state appears. For permalloy films with thicknesses higher than 440 nm, the domain width is of the order of 0.5 μm. The calculated domain width correlates with the stripe domain structure period observed for sputtered permalloy films [50,51]. Note that the domain aspect ratio *a*/*d* for permalloy films under study is not too low and is about 0.4 for a film with a thickness of 1760 nm, as follows from the data presented in Figure 6.

For not-too-narrow domains, when (*a*/*d*)*Q*^1/2^ >> 1, we can obtain an expression for the domain aspect ratio in explicit form. Combining Equations (3) and (10), we have
(13)$$a/d\approx \frac{\pi^{3}}{16.8}\times \frac{k_{A}-H_{a}\cos2\psi /4\pi M_{0}}{1+{\left(1+4\pi M_{0}/H_{a}\right)}^{1/2}}$$

Note that calculations by means of Equation (13) give the same values for the domain width *a*, as shown in Figure 6, when the permalloy film thickness is higher than 220 nm.

The decrease in the Acher parameter with the domain aspect ratio *a*/*d* in films with out-of-plane anisotropy can be explained as follows [17]. With a decrease in *a*/*d*, the dynamic demagnetizing fields at the domain walls increase, which restrains variations in magnetization in the film plane. As a result, magnetization variation in the direction transverse to the film plane becomes more preferable, which leads to a decrease in microwave permeability.

We assume above that there is the stripe domain structure in the permalloy films under study. In general, a more complex micromagnetic configuration may arise in soft magnetic films. However, a comparison of the measurement data on microwave permeability with the results of the model does not contradict the assumption of the existence of a stripe domain structure in the permalloy films.

To conclude this section, note that a decrease in microwave permeability with a growth in film thickness was also studied for cobalt films [19]. For these films, a slow decrease in the resonance frequency and the static permeability was observed, and the Acher parameter reduced gradually from 0.70 to 0.26 when the film thickness increased from 20 to 160 nm. The measured hysteresis loops exhibited a shape close to the transcritical one for cobalt films with a thickness of 470 nm. As in the case of permalloy films, a decrease in the microwave permeability of cobalt films is attributed to the appearance of out-of-plane anisotropy. However, the slower decrease in the Acher parameter in cobalt films is due to the fact that the domain aspect ratio *a*/*d* still remains quite high, even for the thickest films.

## 5. Conclusions

The microwave permeability of a thin film can be estimated based on the saturation magnetization of the material by using the Acher parameter, described in Equation (1). It tends to the limiting value of unity for a single-domain film with in-plane magnetic anisotropy. The appearance of out-of-plane anisotropy and a domain structure results in a decrease in the Acher parameter. The analysis of the Acher parameter can be used to study the relationship between the microwave magnetic properties and magnetic structure of films.

We studied changes in the magnetic structure of permalloy films with an increase in the film thickness based on the measured microwave permeability. The measurement data were analyzed by means of a model for film with a regular stripe domain structure and out-of-plane magnetic anisotropy [17]. The observed decrease in the ferromagnetic resonance frequency with a growth in the film thickness was attributed to the appearance of perpendicular anisotropy. Both the in-plane and perpendicular anisotropy increase with the film thickness due to the magnetoelastic effect. As a result, the static permeability drops sharply when the transition into a transcritical state appears. It should be noted that the ferromagnetic resonance frequency also decreases with a growth in the film thickness, despite the increase in the in-plane anisotropy field. This is due to the fact that the resonance frequency depends on the perpendicular anisotropy field and decreases with the domain aspect ratio *a*/*d* (see Equation (7)).

It is predicted that for soft magnetic films with out-of-plane anisotropy, the value of the Acher parameter is mainly governed by the ratio of the domain width and film thickness. This allows one to estimate the domain width in the film by using measurement data on the microwave magnetic properties. It is found that for relatively thick permalloy films, the domain width is of the order of the film thickness. The formation of a stripe domain structure results in a sharp decrease in the Acher parameter for thick films due to the influence of dynamic demagnetizing fields at the domain walls.

## Figures and Tables

**Figure 1 sensors-24-06165-f001:**
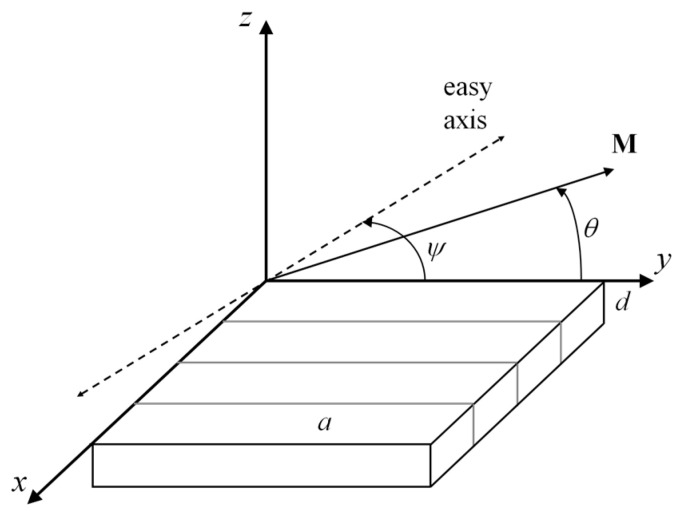
A sketch of the geometry used for analysis. A film of thickness *d* is in the *x*–*y* plane, and the anisotropy axis and equilibrium magnetization vector **M** are in the *y*–*z* plane. The angle between the anisotropy axis and the film plane is *ψ*, and the width of the domains is *a*.

**Figure 2 sensors-24-06165-f002:**
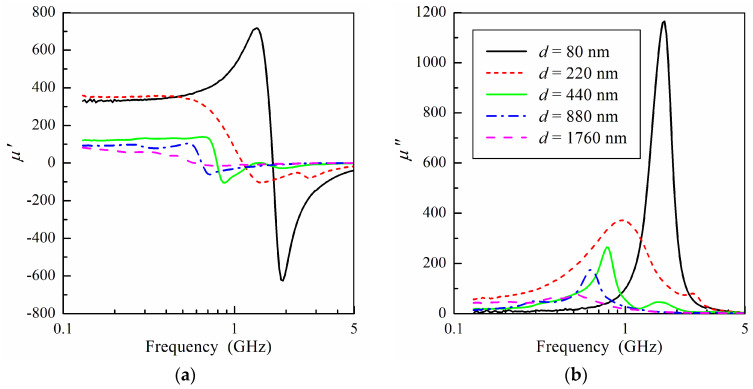
The measured real (**a**) and imaginary (**b**) parts of microwave permeability versus frequency at different values of film thickness *d*.

**Figure 3 sensors-24-06165-f003:**
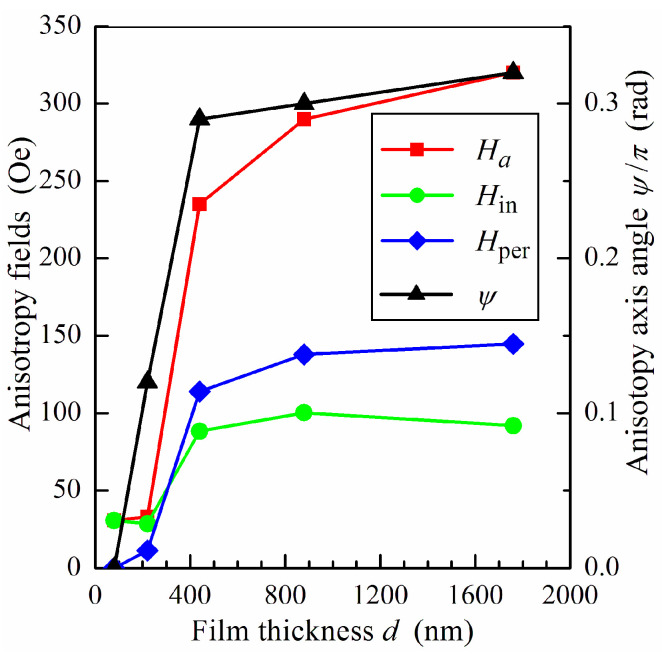
The variations in the anisotropy axis angle *ψ*, effective anisotropy field *H_a_*, in-plane anisotropy field *H*_in_ and perpendicular anisotropy field *H*_per_ (symbols) versus the film thickness *d* used in the analysis. The lines are guides for eyes.

**Figure 4 sensors-24-06165-f004:**
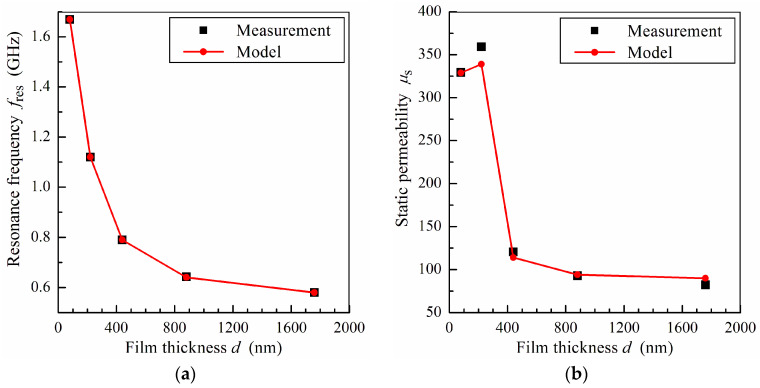
The ferromagnetic resonance frequency *f*_res_ (**a**) and the static permeability *μ*_s_ (**b**) as functions of film thickness *d*. The lines are guides for eyes.

**Figure 5 sensors-24-06165-f005:**
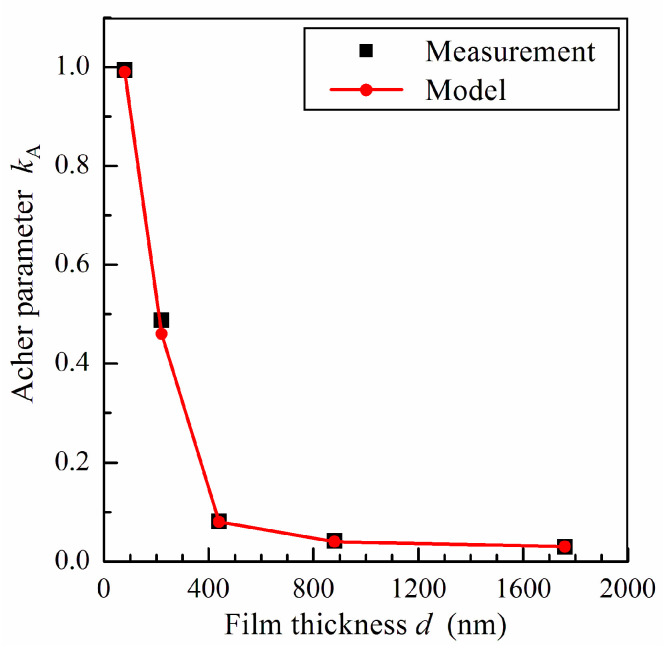
The Acher parameter *k*_A_ as a function of the film thickness *d*. The line is a guide for eyes.

**Figure 6 sensors-24-06165-f006:**
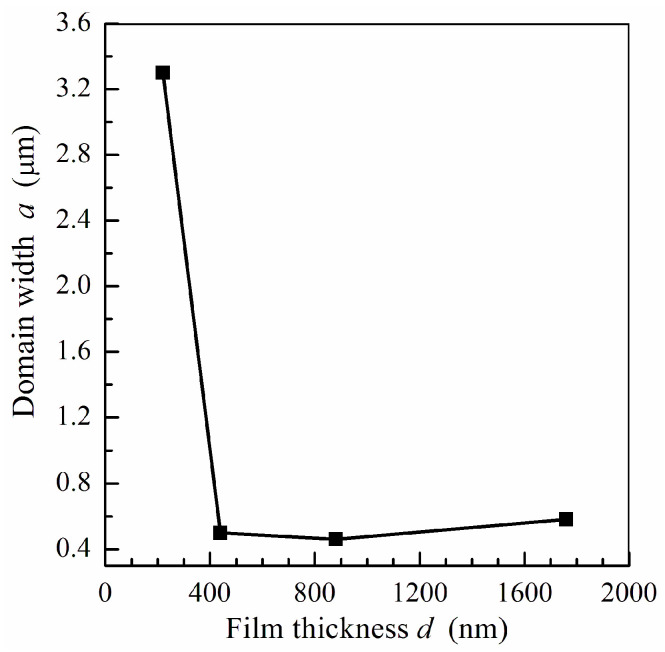
The calculated dependence of the domain width *a* on the film thickness *d* (squares). The line is a guide for eyes.

## Data Availability

Data available from the corresponding author upon reasonable request.

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
