# Peer review of "Analysis of Relationship between Microwave Magnetic Properties and Magnetic Structure of Permalloy Films"

_sensors, 2024, doi:10.3390/s24196165_

Round 1

Reviewer 1 Report

Comments and Suggestions for Authors

The work “Analysis of Relationship between Microwave Magnetic Properties and Magnetic Structure of Permalloy Films” is devoted to the analysis of the correlation between microwave magnetic parameters and the magnetic structure of permalloy films. Using a model previously developed by the authors, the real and imaginary components of magnetic permeability are calculated depending on frequency, as well as the dependence of a number of other magnetic parameters on film thickness (anisotropy fields, static magnetic permeability, Archer parameter and domain width). Despite the fact that the work considers an ideally periodic stripe structure, this should not make a fundamental contribution to the main conclusions and regularities of the work. The manuscript is a coherent, well-structured, understandable and reasonable work. With further development, the results obtained can be useful when considering other objects that have a more complex shape and magnetic structure, for example, amorphous microwires. This work deserves publication in Sensors. I do not see the need to make significant changes to the text of the manuscript.

Reviewer 2 Report

Comments and Suggestions for Authors

The research article "Analysis of relationship between microwave magnetic properties and magnetic structure of permalloy films" discusses the effect of increase in film thickness of static permeability and resonance frequency. The research article is nicely written with some basic equations for the understanding of resonance freq. and static permeability. It also compares experimental results with model calculations. This research article can be published after answering and discussing them in a research article. Especially the mathematical steps involved during calculations mentioned in point (6).     

(1) What is the meaning of static permeability here?   How does the decrease in static permeability is related to increase in plane  anisotropy field? Does the Real Part of Permeability () represent the static permeability?  

(2) What are the equations for "d" dependence of anisotropy in Fig 3?  Like Author plotted Ha , Hin, Hper, and sih in Fif3, showing these fields depends on film thickness, but Eq. 11 and Eq. 12 does not have any 'd' dependence.  

(3) Again in Figure 4, according to Eq. 6 and Eq. 8 does not contain any "d" thickness, how can we write resonance freq. and static permeability in terms of "d"?  

(4) Increase in Film thickness decreases the demagnetization, does that mean that soft magnet acts as single domain magnetic state? Fig. 6 shows that the domain wall width becomes very small, does the author think that the domain wall should disappear?   

(5) What are the studies of thin films in the range of 2nm to 80nm the author should discuss qualitatively in research articles.   

(6) Please write down the steps involved in calculating Eq. 6, Eq. 8? Although the steps are mentioned in Ref 16, it might be useful if the author writes down the steps here especially for Eq. 6 and Eq. 8. The simple micromagnetic way to understand the dispersion relation is very simply discussed in “Atomic and nanoscale spin dynamics”, Journal of Magnetism and Magnetic Materials 502 (2020) 166279 needed to be cited.

Reviewer 3 Report

Comments and Suggestions for Authors

Q1: The references [1-10] should be marked at the corresponding position. The references [29-37] are cited in the same sentence, please state the necessity.

Q2: The last paragraph of the introduction does not highlight the originality of this article. The contributions of this paper and the differences from the literature 16 need to be further described.

Q3: In Figure 4 and Figure 5, the theoretical calculation can include more results than the measured points. In the titles of these pictures, what it the meaning of the sentence “ The line is a guide for eyes”?

Q4: In Figure 6, the results of the domain width should provide the corresponding tested pictures of magnetic domains.

Reviewer 4 Report

Comments and Suggestions for Authors

Manuscript Number: sensors-3164228

Manuscript title: Analysis of Relationship between Microwave Magnetic Properties and Magnetic Structure of Permalloy Films

Journal: Sensors

The relationship between microwave magnetic properties and magnetic structure of permalloy films have been investigated in this manuscript. This manuscript is interesting, well written and easy to follow. The following issues need to be addressed before acceptance.

§  The novelty of the manuscript should be more clearly stated in the abstract of the revised manuscript.

§  The introduction emphasizes the need for high microwave permeability. Are there alternative materials or approaches that have been tested to overcome the limitations in thick films, and how do they compare to permalloy films?

§  Could the properties of the PET substrate influence the observed magnetic properties, particularly at higher thicknesses?

§  The explanation for the sharp drop in permeability with thickness (at 220 nm) is attributed to the appearance of perpendicular anisotropy and stripe domain structures, but this transition is not explained in detail. Why does the transition to perpendicular anisotropy occur at around 220 nm? Is this threshold related to the deposition process, microstructural changes, or intrinsic material properties?

§  The use of the term transcritical state appears several times without a detailed explanation of how this state is identified experimentally in these films. How is the transition into the transcritical state confirmed in these measurements?

§  The magnetoelastic effect and perpendicular magnetic anisotropy should be discussed in more detail, with reported examples (perpendicular magnetic anisotropy at room-temperature in sputtered a-Si/Ni/a-Si layered structure with thick Ni (nickel) layers; perpendicular magnetic anisotropy in multilayers arising from the interplay of thermal strains and diffusion-driven plastic deformation).

§  Could other anisotropy sources, besides the magnetoelastic effect, contribute to the observed changes in magnetic properties at larger thicknesses? Could defects or grain boundaries affect the resonance behavior?

Round 2

Reviewer 3 Report

Comments and Suggestions for Authors

no comments.